

# First-tier detection of intragenomic 16S rRNA gene variation in culturable endophytic bacteria from cacao seeds

Cleiziane Bispo da Silva[1], Hellen Ribeiro Martins dos Santos[1], Phellippe Arthur Santos Marbach[2], Jorge Teodoro de Souza[3], Valter Cruz-Magalhães[1,3], Ronaldo Costa Argôlo-Filho[1] and Leandro Lopes Loguercio[1]

[1] Dept. of Biological Sciences (DCB), State University of Santa Cruz (UESC), Ilhéus-BA, Brazil
[2] Center for Agricultural, Biological and Environmental Sciences (CCAAB), Federal University of Recôncavo da Bahia (UFRB), Cruz das Almas-BA, Brazil
[3] Dept. of Plant Pathology (DFP), Federal University of Lavras (UFLA), Lavras-MG, Brazil

Corresponding authors
Ronaldo Costa Argôlo-Filho,
rcargolofilho@uesc.br,
ronaldoargolo@yahoo.com.br
Leandro Lopes Loguercio,
leandro@uesc.br

## ABSTRACT

**Background**. Intragenomic variability in 16S rDNA is a limiting factor for taxonomic and diversity characterization of Bacteria, and studies on its occurrence in natural/environmental populations are scarce. In this work, direct DNA amplicon sequencing coupled with frequent-cutter restriction analysis allowed detection of intragenomic 16S rDNA variation in culturable endophytic bacteria from cacao seeds in a fast and attractive manner.

**Methods**. Total genomic DNA from 65 bacterial strains was extracted and the 16S rDNA hyper variable V5–V9 regions were amplified for enzyme digestion and direct Sanger-type sequencing. The resulting electropherograms were visually inspected and compared to the corresponding *Alu*I-restriction profiles, as well as to complete genome sequences in databases. Restriction analysis were employed to substitute the need of amplicon cloning and re-sequencing. A specifically improved polyacrylamide-gradient electrophoresis allowed to resolve 5-bp differences in restriction fragment sizes. Chi-square analysis on $2 \times 2$ contingency table tested for the independence between the 'number of *Alu*I bands' and 'type of eletropherogram'.

**Results**. Two types of electropherograms were obtained: unique template, with single peaks per base (clean chromatograms), and heterogeneous template, with various levels of multiple peaks per base (mixed chromatograms). Statistics revealed significant interaction between number of restriction fragments and type of electropherogram for the same amplicons: clean or mixed ones associated to ≤5 or ≥6 bands, respectively. The mixed-template pattern combined with the *Alu*I-restriction profiles indicated a high proportion of 49% of the culturable endophytes from a tropical environment showing evidence of intragenomic 16S rDNA heterogeneity.

**Conclusion**. The approach presented here was useful for a rapid, first-tier detection of intragenomic variation in culturable isolates, which can be applied in studies of other natural populations; a preliminary view of intragenomic heterogeneity levels can complement culture-dependent and -independent methods. Consequences of these findings in taxonomic and diversity studies in complex bacterial communities are discussed.

## INTRODUCTION

Correct taxonomic identification and proper estimates of bacterial diversity are very important issues, due to wide environmental distribution, ecological functions, pathogenic potential and biotechnological applications of this domain. For more than a century, the identification/classification of bacteria has been done exclusively through culture-dependent methods, based on a series of morphological, physiological and biochemical tests after growth in appropriate media (*Janda & Abbott, 2002*). With the advances in molecular genetics, a criterion of DNA-DNA hybridization (DDH) showing a reassociation of 70% or more was established to define genospecies (*Stackebrandt & Goebel, 1994*). In the genomic era, various concepts and approaches have been debated as biologically meaningful systems for bacterial species definition (*Konstantinidis, Ramette & Tiedje, 2006*).

Sequencing of the 16S ribosomal RNA gene in prokaryotes has been widely used to determine taxonomic and phylogenetic relationships (*Clarridge, 2004*; *Armougom & Raoult, 2009*; *Manaka, Tokue & Murakami, 2017*). Bacterial strains with ≥70% DNA-DNA reassociation usually have >97% identity in their 16S rRNA gene sequence; on the other hand, less than 70% DNA-DNA hybridization, even with almost identical 16S rDNAs, may indicate different species (*Janda & Abbott, 2007*). This is especially relevant when ecological niches are included in the comparative analyses (*Gevers et al., 2005*; *Konstantinidis, Ramette & Tiedje, 2006*). Multi-locus sequencing analysis (MLSA) is also used for prokaryotic species definition, counting on a set of specific genes with sufficient evolutionary signals to allow fine discrimination between genetically close strains (*Gevers et al., 2005*). Polyphasic approaches (*Vandamme et al., 1996*) including all these methods have been regarded as the most suitable manner for taxonomic characterization of bacteria (*Das et al., 2014*; *Sarethy, Pan & Danquah, 2014*).

An often overlooked aspect to consider in the analyses of newly isolated strains from environmental samples (including endophytes) is how they are preliminarily assigned to a taxon (*Woo et al., 2011*). Studies have shown that horizontal gene transfer and recombination within the 16S rDNA do occur in bacteria (*Schouls, Schot & Jacobs, 2003*; *Zhaxybayeva et al., 2006*; *Kitahara & Miyazaki, 2013*; *Tian et al., 2015*). Experimental transfer of complete rRNA operons within and between species have resulted in viable organisms with heterologous 16S rRNA genes from two other species (*Asai et al., 1999*). Since some strains can carry up to 15 copies of the 16S rRNA gene (*Klappenbach et al., 2001*; *López-López et al., 2007*; *Engene & Gerwick, 2011*; *Sun et al., 2013*), transfer and recombination of segments can lead to a mosaic-like structure, i.e., different sequences of the 16S rDNA within the same cell (*Eardly, Wang & Van Berkum, 1996*; *Schouls, Schot & Jacobs, 2003*; *Tian et al., 2015*). The presence of multiple 16S rDNA copies and possible intragenomic heterogeneity can be, therefore, a limiting factor for both correct identification and counting of operational taxonomic units (OTUs) in biodiversity studies

of both culturable and unculturable bacteria (*Coenye & Vandamme, 2003*; *Pei et al., 2010*; *Chen et al., 2015*). Although culturable microbes represent only 0.1–5% of the estimated total microbial diversity of any given environment (*Bull, 2004*), the levels of 16S rDNA intragenomic variation in a culturable population may provide a glance of what can occur in the whole community. Knowing such variation is useful in studies on microbial ecology and diversity, to prevent overestimation of this latter parameter (*Sun et al., 2013*) and to provide correction factors to compensate it.

Due to an intimate interaction with the plants, endophytic microbial symbionts can play important roles in host adaptation and evolution (e.g., *Bulgarelli et al., 2013*; *Turner, James & Poole, 2013*; *Agler et al., 2016*), performing relevant biological functions, such as increased photosynthetic efficiency, growth promotion, tolerance to abiotic and resistance to biotic stresses, including antagonism towards phytopathogens (*Barrow et al., 2008*; *Hanada et al., 2009*; *Andreote, Gumiere & Durrer, 2014*; *Hardoim et al., 2015*; *Berg et al., 2017*). Endophytes also represent a potential source of bioactive compounds with a variety of applications in agriculture and industry (*Schulz et al., 2002*; *Schulz & Boyle, 2005*; *Leite et al., 2013*; *Gouda et al., 2016*). The great genetic, molecular and biochemical diversification in microorganisms is the basis of these processes, with their greatest biodiversity being long acknowledged in tropical latitudes (*Strobel & Daisy, 2003*; *Arnold & Lutzoni, 2007*; *Duarte et al., 2013*). The cacao tree is an interesting experimental model for these regions, due to a globally recognized importance for its economic, social, and environmental characteristics (*Donald, 2004*; *Schroth et al., 2011*; *Beg et al., 2017*; *Wickramasuriya & Dunwell, 2018*). Moreover, exploratory studies on tropical cacao communities of endophytic fungi and bacteria in different plant tissues have been conducted, with emphasis on improvement of the plant productivity based on biological control and growth promotion (*Rubini et al., 2005*; *Crozier et al., 2006*; *Mejía et al., 2008*; *Hanada et al., 2009*; *Hanada et al., 2010*; *Melnick et al., 2011*; *Leite et al., 2013*; *Tchinda et al., 2016*).

Mixed electropherograms obtained from Sanger sequencing can be an interesting first-tier approach to assess polymicrobial samples composition or intragenomic heterogeneity of rRNA genes/operons. Previous studies have demonstrated the feasibility of employing direct sequencing of PCR-amplified 16S rDNA stretches to detect the presence of multiple microbes (*Kommedal, Karlsen & Sæbø, 2008*; *Hartmeier & Justesen, 2010*) or intragenomic rDNA variability in clinical samples of culturable isolates (*Chen et al., 2015*) by visually inspecting the corresponding mixed chromatograms. In the present work, a similar approach of Sanger-type chromatogram-based assessment, but complemented with frequent-cutter (*Alu*I) restriction analyses was applied to study intragenomic variability in 16S rDNA in a set of culturable endophytic bacteria from cacao seeds. The results confirmed detection of intragenomic variation in individual isolates, suggesting this feature as being present in natural/environmental populations at high frequencies.

## MATERIALS & METHODS

### Bacterial strains collection

The 65 endophytic bacterial isolates used in this work belonged to the Laboratory of Agroindustry Applied Microbiology of the State University of Santa Cruz (LABMA/UESC,

Ilhéus-BA, Brazil), and were previously obtained from pulp adhered to seeds from cacao pods (*Da Silva, 2013*). The isolates were purified by the single-colony streak-plate method in Nutrient Agar and Tryptone Soy Agar, and maintained at 30 °C in the dark. This single-colony streaking and culturing procedure was repeated at least three times to assure that only homogeneous/pure colonies were obtained. The total DNA from the isolates was extracted by the *Doyle & Doyle (1987)* method.

## Amplification and sequencing of the 16S rRNA genes from culturable isolates

Amplification of the V5–V9 hypervariable region of the 16S rRNA gene from the bacteria was performed by PCR with the primer-pair 799F (5′-AACMGGATTAGATACCCKG-3′) and U1492R (5′-GGTTACCTTGTTACGACTT-3′) (*Chelius & Triplett, 2001*). Each 25-$\mu$L (final volume) of polymerase chain reaction contained 8 ng of extracted DNA template, 2.5 $\mu$L of 10x *Taq* buffer, 1.25 $\mu$L of 50 mM $MgCl_2$, 2.5 $\mu$L of 2 mM dNTP, 0.2 $\mu$L of Platinum® *Taq* DNA polymerase (5 U $\mu L^{-1}$) (Invitrogen$^{TM}$), 15 pmoles of 799F, 7.5 pmoles of U1492R, and 0.25 $\mu$L BSA at 0.1%. The reaction was performed under the following conditions: 3 min at 96 °C, followed by 30 cycles of 20 s at 94 °C, 40 s at 58 °C and 40 s at 72 °C, with a final extension step at 72 °C for 10 min. Aliquots of 5 $\mu$L of each reaction were analyzed on 1% (w/v) agarose gel electrophoresis in TBE buffer.

The amplified fragments were purified from agarose gels using *PureLink® Quick Gel Extraction Kit* (Invitrogen$^{TM}$), following manufacturer's recommendations. The purified DNA from gel were quantified by the *NanoDrop ND-1000* spectrophotomer (Thermo Scientific$^{TM}$) prior to sequencing. The gel-purified amplicons were sequenced through the ABI-PRISM® 3100 Genetic Analyzer System, equipped with 50-cm capillaries and POP6 polymer. For each sequencing reaction, 3 $\mu$L of the BigDye$^{TM}$ Terminator v3.1 Cycle Sequencing RR-100 reagent was used, with DNA template at ∼50 ng, and 2.5 pmoles of the 799F primer, in a final volume of 10 $\mu$L. The sequencing reactions were done in GeneAmp® PCR System 9700 thermocycler under the following conditions: 3 min at 96 °C, followed by 25 cycles of 10 s at 96 °C, 5 s at 55 °C and 4 min at 60 °C. The reaction products were precipitated with 4x their volume with 75% isopropanol for 30 min, centrifuged at 13 krpm for 15 min; the pellet was washed with 60% ethanol and dried to completion. Subsequently, the pellets were diluted in 10 $\mu$L of Hi-Di formamide, denatured at 95 °C for 5 min and cooled on ice also for 5 min, prior to being electro-injected in the automatic sequencer. The sequencing data were collected by the Data Collection v 1.0.1 program with the following parameters: Dye Set 'Z', Mobility File 'DT3100POP6 {BDv3} v1.mob', Run Module 1'StdSeq50_POP6_50 cm_cfv_100', and analysis Module 1 'BC 3100SR_Seq_FASTA.saz'.

## Electropherograms and restriction analyses of the 16S rRNA gene sequences

A visual inspection of the resulting electropherograms was carried out, including analysis of peaks definition and intensity, the presence of overlapping peaks and extension of the overlapping stretches. For a preliminary identification of isolates, the final base-called, processed sequences (through the ABI Sequencing Analysis application) obtained

from the 16S rRNA genes of the endophytic bacterial isolates were submitted to the GenBank by the BlastN software (http://www.ncbi.nlm.nih.gov/BLAST/). Based on the list of genera/species obtained from the BlastN, a further search was done for those with complete genomes deposited in the database. For quality and safety of information from the BlastN search, only those sequences obtained from clean, single-peaks chromatograms of the bacterial endophytic isolates from cacao (see above) were aligned with the complete genome sequences of the corresponding species (see Results). A supplementary multi-fasta text file containing these sequences is provided, with the isolates ordered according to levels of identity retrieved from BlastN (Table 1). In addition, all the corresponding electropherograms, labelled with the corresponding isolate identification, were provided as a compressed file (*.zip) as Supplementary Information.

The same PCR products subjected to sequencing were also digested with *Alu*I (AG/CT) restriction enzyme (Uniscience® do Brasil) in reactions composed of 0.8 µL of 10x enzyme buffer, 0.25 µL of the *Alu*I enzyme (10 U µL$^{-1}$), 2 µL of the PCR reaction, brought up to final volume of 8 µL with ultra-pure water. The *Alu*I-digestion reactions were incubated in a water bath at 37 °C for 50 min, following the enzyme manufacturer's recommendations. This 4-bp frequent cutter was chosen because it yielded sufficiently discriminatory restriction profiles after electrophoresis for the amplified V5–V9 16S rRNA gene regions from the bacterial isolates (*Dos Santos, 2017*).

For the separation of the *Alu*I digestion products, a previously defined procedure with a high-resolution ability (*Dos Santos, 2017*) was used for the analysis. The *Alu*I digestions were submitted to vertical electrophoresis in 5–11% polyacrylamide (w/v) gradient gel in 1x TAE buffer (20 mM Tris-acetate, 0.5 mM EDTA, pH 8) at 80 V for 16 h. The gels were stained for 30 min in the dark in a solution composed of 15 µL of GelGreen$^{TM}$ for each 50 mL of distilled water (3:10$^4$ ratio). After the staining time, the gels were photodocumented in Blue LED Transilluminator (Nippon Genetics Europe). The gel images were analyzed for counting of the restriction fragments generated; this procedure allowed the unambiguous detection of individual fragments with a size-difference equal to, or greater than 5 bp. Additionally, *in silico* analyses of the endophytic bacterial sequences with clean, single-peaks chromatograms was performed to locate *Alu*I cleavage sites and predict the number of fragments to be generated after the restriction digestions.

Each isolate's 16S rDNA was subjected at least twice to the whole procedure of PCR amplification, sequencing and enzyme digestion, thereby composing a minimum of two biological replicates per sequence obtained; the results were consistent among replicates.

## Statistics

A non-parametric chi-square analysis was performed in a 2 × 2 contingency table, with correction of Yates, to test the null hypothesis of independence between the variation factors, i.e., "type of chromatogram" ('clean' and 'mixed') vs "number of *Alu*I fragments" ('up to 5 bands' and '6 or more bands'). These two categories of *Alu*I-generated fragments were set according to the average number of restriction sites per single 16S rDNA sequence. After assessing 143 16S rDNA sequences of endophytic bacteria from databases, we found that 82% of them have 2 to 4 *Alu*I sites in the V5–V9 region, which generates 3 to 5

**Table 1** Approximate identification of culturable endophytic bacterial isolates from cacao, based on direct amplicon sequencing of the 16S rDNA V5–V9 hypervariable region.

| Electrophero-gram[a] | Isolate[b] | BlastN results[c] (sp/strain) | Identity (%) |
|---|---|---|---|
| Clean(15) | 5, 6, 18, 31 | *Bacillus cereus* LV11 (KU705859.1), *B. thuringiensis* strain NBRC 101235 (NR112780.1), *B. pumilus* strain L1 (KT937148.1), *B. safensis* strain IHB B 14105 (KM817280.1) | 100 |
| | 3, 11, 23, 34, 37, 38, 39 | *Bacillus amyloliquefaciens* strain HD34 (KT368090.1), *B. pumilus* strain AUCAB16 (JN315777.1), *B. pumilus* strain NBRC 12092 (NR112637.1), *Bacillus* sp. BAB-4112 (KJ778656.1), SMF5 (AJ868359.1), SW3.2 (KU740234.1), *B. stratosphericus* strain IHB B 6832 (KF668462.1) | 99 |
| | 53, 57, 62, 67 | *Gluconobacter nephelii* strain LMG 26773 (NR118638.1), *Lysinibacillus fusiformis* strain NBRC 15717 (NR112628.1), *Raoultella ornithinolytica* B6 (CP004142.1), *Staphylococcus pasteuri* strain ATCC 51129 (NR024669.1) | |
| Mixed (50) | 40, 63 | *Bacillus stratosphericus* strain IHBB 9411 (KR085786.1), *Staphylococcus epidermidis* strain DAR1907 (CP013943.1) | 99 |
| | 52 | *Escherichia coli* strain BAB-538 (KF535120.1) | 98 |
| | 17, 54 | *Bacillus pumilus* strain IHB B 12534 (KJ767390.1), *Lelliottia amnigena* strain ZB04 (CP015774.1) | 96 |
| | 36; 13, 10, 61; 1, 29, 48, 51; 25, 43, 59, 71; 8, 24, 27, 46, 58, 69; 33, 44, 64, 68; 12, 30, 42, 55 | *Bacillus* sp. SB3.1 (KU740223.1); *Bacillus australimaris* strain MCCC 1A05787 (NR148787.1), *B. pumilus* (KU922935.1), *Pseudomonas plecoglossicida* strain RD_AZLTR_14 (KU597542.1); *B. altitudinis* strain BT 98 (KJ848598.1), *B. safensis* strain AL-8 (HQ848126.1), *Citrobacter* sp. BRRO1 (KT735246.1), *Enterobacter sp.* clone HSL29 (HM461152.1); *B. pumilus* strain SQU P001 (KU220846.1), *B. subtilis* strain SRI2 (KP271983.1), *Pantoea agglomerans* (DQ392984.1), Uncultured bacterium clone 218002-244 (JQ940965.1); *B. licheniformis* strain KYLS-CU01 (KF111800.1), *B. pumilus* strain SH-B9 (CP011007.1), *B. pumilus* strain TP-Snow-C22 (HQ327131.1), *Brevibacillus* sp. XYY-2015 (KR528483.1), *Paenibacillus* sp. KMSDS2 (JF768723.1), *Staphylococcus lugdunensis* strain ATCC 43809 (NR024668.1); *Bacillus* sp. 01082 (EU520309.1), *B. subtilis* strain W1 (KC441816.1), *Staphylococcus saprophyticus* strain PW64 (KT726989.1), *S. warneri* strain LEH1_5A (JN644590.1); *Bacillus pumilus* strain C1C5502 (KR677555.1), *B. safensis* strain FFA38 (JN092820.1), *B. subtilis* strain SP3 (KT875349.1), *Lysinibacillus fusiformis* strain BN-4 (JN039176.1) | 86–80 |
| | 2, 60, 66, 70, 72; 47, 49, 50; 26; 45; 32, 41 ; 28 ; 4, 9 ; 7, 16, 35, 56 | *Bacillus altitudinis* strain JF2 (KC171985.1), *Pantoea agglomerans* strain A9 (KC434965.1), *Staphylococcus* sp. CM2E1 (KM874434.1), *S. warneri* strain MBS022 (KT582294.1), Uncultured *Lactobacillales bacterium* isolate DGGE 6PLAB (GQ911039.1); *Citrobacter murliniae* strain E61 (HQ407238.1), *Enterobacter asburiae* strain RCB875 (KT261087.1), *Enterobacter* sp. A7 16S (JX081588.1); *Bacillus pumilus* strain SW-3 (KC813157.1); *Bacillus thuringiensis* strain Po-5 (JX391979.1); *Bacillus safensis* strain MUGA141 (KJ672329.1), *B. subtilis* strain p95_H01 (JQ830651.1); *Bacillus pumilus* strain ZK1 (JQ773350.1); *Bacillus atrophaeus* strain NBF1 (HQ256518.1), *B. pumilus* (GQ861537.1); *Bacillus firmus* (EF526504.1), *B. pumilus* strain IARI-SL-5 (JX645203.1), *Bacillus* sp. Ob 06 (AJ971891.1), *Lysinibacillus* sp. TRS6 (KJ617407.1) | 79–70 |

**Notes.**

[a]An expedite characterization of each culturable endophytic isolate from cacao was performed through direct amplicon sequencing on gel-purified, single-band PCR products of 799F/1492U primers, spanning the V5–V9 hypervariable region of 16S rDNA. Analysis of the resulting electropherograms indicated two major groups: one with clear, sharp and undoubtedly-single peaks for each nucleotide (high quality sequences), which comprised 15 isolates, within which 93.3% expectedly showed 3–5 *Alu*I restriction fragments (see Results and Discussion). The other group (mixed-peaks) presented variable levels of background, lower-intensity peaks underneath each nucleotide-read peak (lower quality sequences); this group comprised 50 isolates, within which 64.0% expectedly showed ≥6 *Alu*I restriction fragments (also see next).

[b]Numbers correspond to the identification of the culturable isolates in the local collection; underlined isolates are those whose *Alu*I restriction patterns departed from the expected number of bands, i.e. ≥6 fragments for the single-peaks group and ≤5 fragments for the mixed-peaks group (see chi-square analysis in the Results text).

[c]Considering the highest score obtained from the BlastN search (with 100% sequence cover), the species/strains retrieved are shown by the corresponding access number indicated between parenthesis. The lower levels of identity (≤98%) indicated in the right column indicate an increasing interference of the lower-intensity underneath peaks, which generate a "chimeric-sequence" effect in the main base-called reads, thereby departing from the expected (99–100%) sequence similarity.

restriction fragments. Therefore, if the PCR amplicons are composed of more than a single type of template, then the number of *Alu*I-bands generated after digestion have ∼82% probability to be equal or higher than six. A very significant chi-square value ($P < 0.01$) indicates association between the type of chromatogram and the number of bands detected in the high-resolution polyacrylamide-gradient electrophoresis (*Dos Santos, 2017*), so that, clean (single-peaks) chromatograms tend to associate with ≤ 5 bands, whereas mixed chromatograms correlates well with ≥ 6 restriction fragments (see Results).

## RESULTS

A total of 65 culturable endophytic isolates obtained from cacao pulp adhered to seeds (*Da Silva, 2013*) were considered in this study. Their DNA was individually extracted and subjected to PCR amplification with specific primers targeting the V5–V9 hypervariable region of the 16S rDNA, 799F and U1492R (*Chelius & Triplett, 2001*). Each isolate generated a single PCR amplicon (electrophoretic band) that were individually sequenced in a direct manner (Sanger method), after purification from the gel. Under these conditions, no apparent unspecific amplification was observed in the gel for all samples. The preliminary identification of isolates, patterns of chromatogram obtained, and levels of sequence identity revealed by the BlastN are summarized in Table 1. The electropherograms from the 16S rDNA sequences presented two major patterns (Fig. 1): 15 isolates showed clear, single-peaks electropherograms of high quality sequences ('clean'), whereas 50 presented mixed eletropherograms, with variable number and intensity of peaks underneath the main base-called sequences subjected to BlastN (Table 1). When the *Alu*I restriction analysis was performed on the same PCR-amplified and gel-purified amplicons from the endophytic isolates, all but one of the clean sequences (93.3%, Table 1) presented the expected electrophoretic pattern of fragments, according to the number of *Alu*I sites identified *in silico* for those sequences (Fig. 1).

*In silico* analysis for the *Alu*I sites of the mixed electropherograms could not be done safely, due to the uncertainty of their base-called sequences (Table 1); hence, the number of restriction fragments generated was assessed by electrophoresis only. For these 50 mixed sequences, the majority of *Alu*I restriction patterns (63.3%) showed ≥6 bands (Fig. 1). A chi-square analysis ($2 \times 2$ contingency table) was performed to test the null hypothesis ($H_o$) of independence between the 'number of *Alu*I fragments' (≤5 or ≥6 bands) and the 'type of sequence' (single or mixed). The results ($\chi^2 = 12.968$; $P = 0.0003$) allowed to reject $H_o$, thereby indicating that the number of *Alu*I restriction fragments was essentially dependent upon the type of chromatogram obtained. Interestingly, in certain cases of mixed sequences, possible *indels* in the 16S rRNA genes have apparently generated mixed templates detectable from specific points in the electropherogram (Fig. 2) (*Chen et al., 2015*). From a visual inspection and alignment, *indels* could be identified between the main and the underneath sequences, although the nucleotide signals for the latter were of such a lower level that further fluorescence-background interference prevented their safe detection. In the example shown (Fig. 2), the main base-called sequence was 99% identical to a *Bacillus* strain, whereas the underneath sequence dropped the identity to 96% closest to

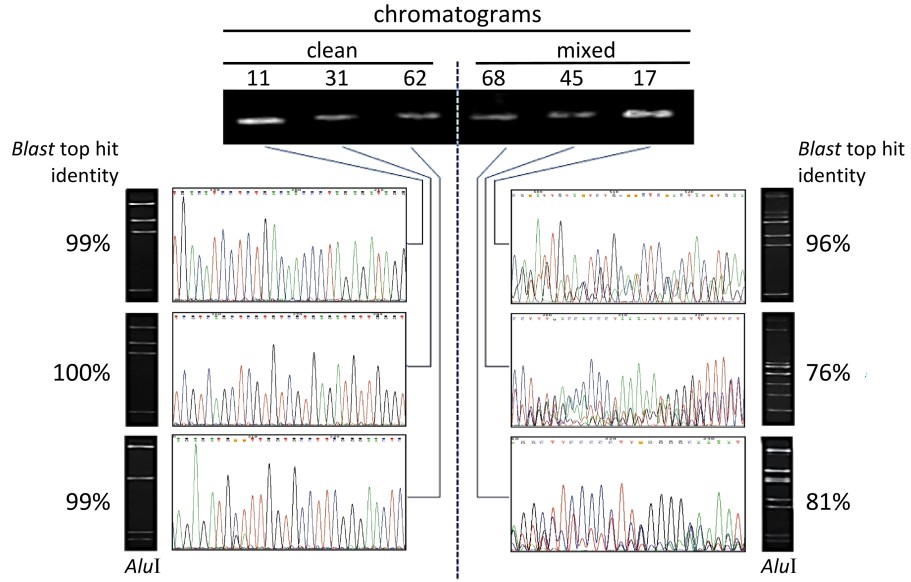

**Figure 1** **Illustration of electropherograms and restriction profiles resulting from sequencing and** ***Alu*I digestion of PCR-amplified V5–V9 region of 16S rRNA genes from endophytic bacteria from** ***Theobroma cacao.*** The procedure was applied to 65 culturable isolates from cacao seeds + pulp (see text), and the data correspond to six isolates (identified by the numbers on top) that are representative examples. The agarose-gel electrophoretic profiles for single-fragment PCR-amplifications with primers 799F/U1492R are shown on the top. 'Clean' electropherograms, characteristic of unique templates, are shown on the left side, whereas mixed ones, which can indicate multiple templates, appear on the right side. The images of the corresponding gel resulting from the *Alu*I digestion of the same sequence appear next to each chromatogram. The percentage of identity with the best hit retrieved by the BlastN is indicated besides each gel. (The sequences produced from the 65 isolates are provided in the Supplementary Information).

a *Paenibacillus* strain. These results strongly indicated the presence of intracell heterogeneity for the 16S rDNA for that particular isolate. Obviously, the observation of clean or mixed chromatograms (which pointed to unique or multiple templates, respectively) did not allow any inference about the number of copies of the 16S rRNA gene present in the organism.

To verify whether the strains retrieved by the BlastN search also show evidence of different 16S rRNA genes in fully sequenced genomes of the same species, an *in silico* analysis was performed with the 15 clean and the two mixed sequences showing 99% identity to database entries (see Table 1). These sequences were chosen due to their safer taxonomic identification, resulting from clearer and undoubtful base-called sequences that provided the highest cover and identity percentages when aligned with GenBank entries (Table 1, Fig. 1). From those 17 safe sequences, we were able to identify 11 deposited strains with fully sequenced genomes from four bacterial species: *Bacillus thuringiensis*, *B. pumilus*, *Gluconobacter oxydans* and *Staphylococcus epidermidis* (Fig. 3). The results showed a range of one to seven aligned copies of 16S rRNA gene that were different from each other and from our isolates. Most of the variations were base substitutions, but deletions and duplications were also identified (Fig. 3). Interestingly, a copy of 16S rRNA gene within

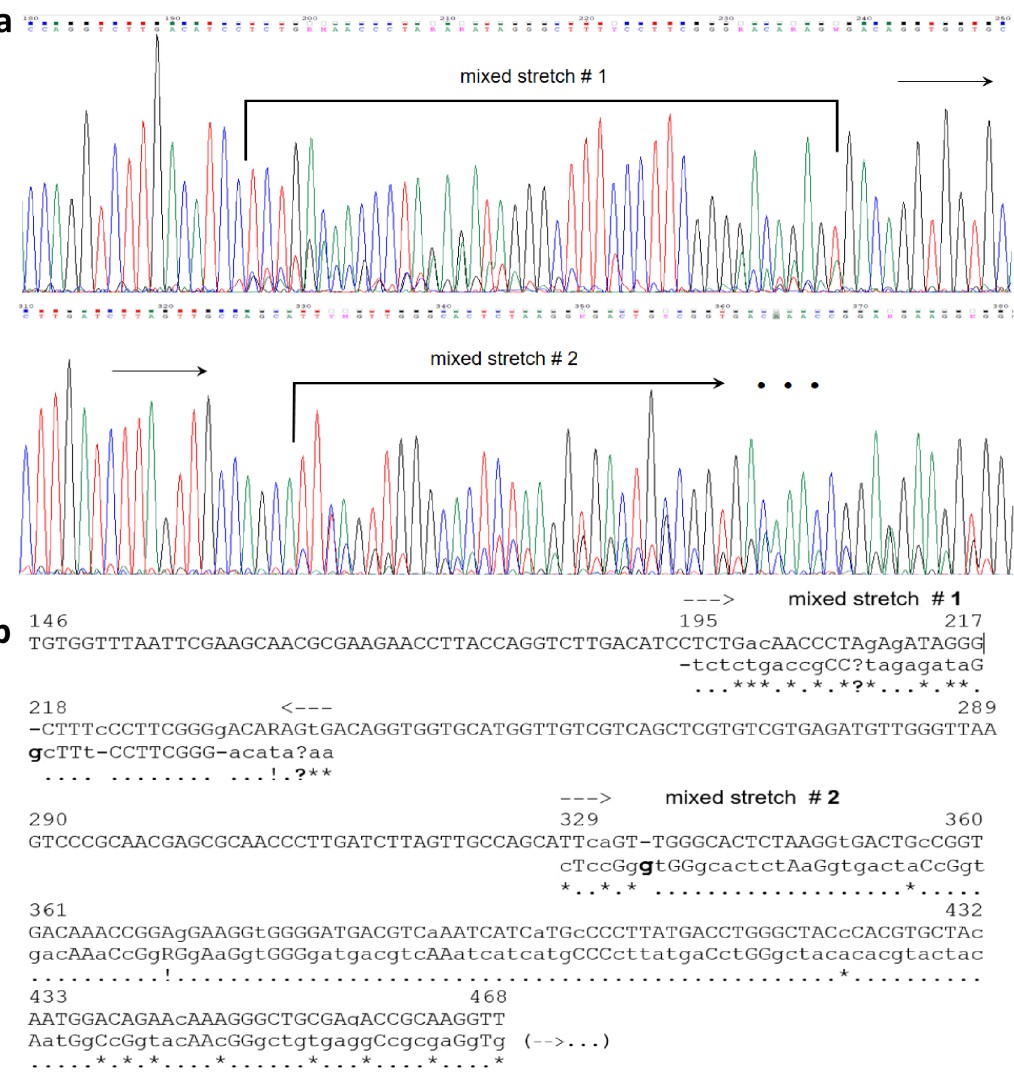

**Figure 2 Example of a double-template mixed electropherogram for a 16S rRNA gene amplicon of an endophytic bacterial isolate from *Theobroma cacao*.** The region displayed shows specific stretches where a clean pattern changes to a double-sequence electropherogram, indicated by horizontal arrows. The base-called sequence of the respective 16S rRNA gene are given on top of the electropherogram (A), and as the top sequence (capital letters) in the underneath alignment. Visual inspection of peak intensities allowed precise identification of the main and secondary sequences, which were manually aligned as indicated (B). For the stretches where only single peaks were observed, the sequences of the supposed two templates appeared to be the same and are shown as capital letters in a single line. For the aligned 'mixed' stretches 1 and 2, the top sequence is the main one and the borders of the overlapping stretches are indicated by dashed arrows and the base-pair number (position) of the main sequence; small-caps letters in the main (top) sequence were base-called ambiguities defined by visual inspection of the eletropherogram. The secondary sequence was predominantly shown in small-caps, except for those bases with only a single peak (capital letters). The two sequences alignment was optimized manually, in which dashes were introduced in each sequence for the best possible adjustment. Dots and stars underneath the alignment correspond to matches and mismatches, respectively; question marks indicate points were the secondary-sequence signals were too low and it was not possible to discern them from background; bold 'g' letters and dashes suggest potential *indels* between the two sequences. The sequences definition and alignment could be undoubtedly done up to the 468 bp only.

the genome of the *B. pumilus* strain CP009108.1 showed an insertion of 14 bp not shared by the other seven copies, nor by our isolate '23', which made it only 97% identical to them (Fig. 3B). It is also noteworthy that, for most of the clean-chromatogram's isolates subjected to BlastN, the corresponding list of hits depicted from 7 to 51 different species with nucleotide identities ≥98%, thus providing a series of distinct genera/spp as possible candidates for taxonomic definition of an isolate, based on this 16S rDNA region.

## DISCUSSION

Sequences of 16S rRNA gene have long been used for identification of prokaryotic organisms (*Srinivasan et al., 2015*), at least in a preliminary manner. However, the intragenomic variation naturally present for this gene, both quantitatively (number of copies) and qualitatively (different DNA sequences), has been adding complexity to this matter (*Coenye & Vandamme, 2003*; *Sun et al., 2013*; *Chen et al., 2015*; *Valdivia-Anistro et al., 2016*). Due to the extensive use of 16S rRNA gene sequences for taxonomy, phylogenetics and metagenomics, for both culture-dependent and -independent methods, the assessment of the extent of these heterogeneities in natural populations appears as an important issue (*Pei et al., 2010*; *Michon et al., 2010*). Knowing the levels of intragenomic variation (i) in individual bacterial strains, (ii) in a genus or species, or (iii) in more diverse ecosystems (e.g., water, soils, plants, animals and humans' microbiomes) would help researchers to adjust biodiversity estimates of populations/communities based on 16S rDNA sequences, as well as to choose which rDNA region is more appropriate to use for taxonomic identification purposes (*Sun et al., 2013*; *Chen et al., 2015*). In the present study, previously isolated endophytic bacteria from cacao seeds (*Da Silva, 2013*; *Dos Santos, 2017*) were used as a model system to address levels of intragenomic variation of this proxy gene in a natural population from a tropical environment. The assessment of an electropherogram profile from direct Sanger sequencing of rDNA-amplified fragment from a single culturable bacterial isolate, coupled with a frequent-cutter (*Alu*I) restriction analysis of the amplicon, could serve properly as a first-tier indication of intragenomic heterogeneity in the 16S rRNA gene. In this case, the frequent-cutter restriction analysis was able to properly replace the need of cloning and re-sequencing the PCR amplicons (e.g., *Chen et al., 2015*) to confirm the multiple template condition.

The results obtained from the combined analyses of electropherograms and *Alu*I restriction patterns (Fig. 1, Table 1) showed that clean chromatograms tend to be indicative of a unique template (with the number of restriction fragments in electrophoresis agreeing with the number of *Alu*I sites seen *in silico*), whereas the mixed ones indicate more than a single template (frequently with ≥ 6 *Alu*I bands) present in the sequenced amplicon. These results, which were fully supported by the chi-square analysis, is in close agreement with previous work with polymicrobial (*Kommedal, Karlsen & Sæbø, 2008*) and with clinical strictly anaerobical samples (*Chen et al., 2015*). The number of underneath peaks varied among mixed sequences, which may correspond to the number of templates available (only one underneath peak suggests two distinct templates; two underneath peaks, three templates, and so on; *Kommedal, Karlsen & Sæbø, 2008*).

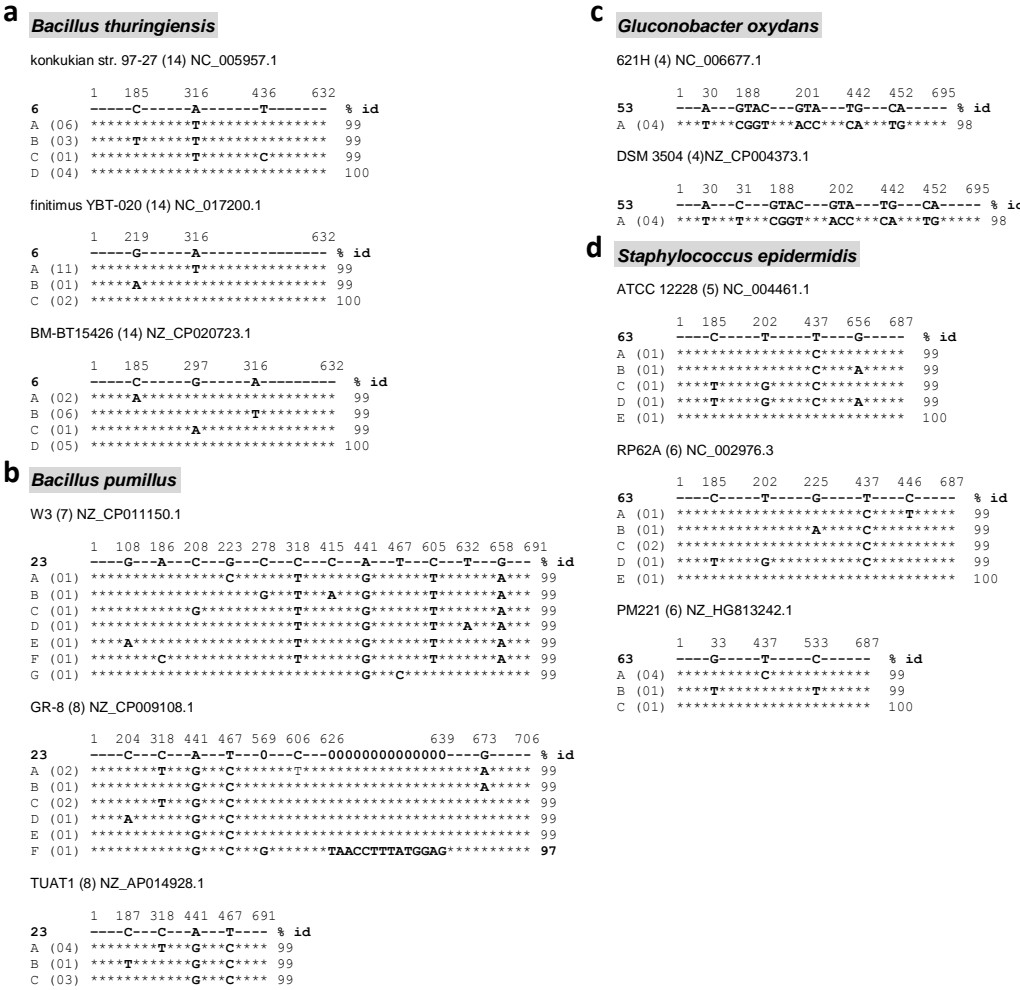

**Figure 3** **Intragenomic variation of the 16S rRNA gene from bacterial species with their complete genome deposited at GenBank, identified by the BlastN search with sequences from endophytic isolates from *Theobroma cacao*.** Only the 15 undoubtful, clean sequences and the two mixed sequences with 99% identity (Table 1) were considered. The indicated genomes (species names in grey boxes) were found by BlastN with the isolates # 6 (A), 23 (B), 53 (C) and 63 (D), which were indicated in bold as the reference sequence used in each of the 11 alignments shown. The identification and access numbers of the corresponding strains with genomes fully sequenced is indicated on top of each alignment group. In each of these groups, the letters A, B, C etc on the left column correspond to the different types of 16S rRNA gene sequences found in the genome, with the number of copies of each type shown between parenthesis besides them. The numbers in the first line (out of scale) correspond to the base-pair positions of the cacao isolate sequence where differences were found among the sequence types. The asterisk (*) sign indicate bp matches in the whole stretches compared to the reference sequence (cacao isolate) on top of each alignment group. The percentage of sequences identity with respect to the cacao isolates is shown on the right side of each sequence. For the second alignment group of *Bacillus pumilus* (isolate 23), the stretch of 'zeros' on the reference sequence indicate a large gap in relation to the 'F' sequence (the gap is present in all sequence types, from 'A' to 'E').

The generally much lower identities (≤96%) obtained from 47 mixed sequences (Table 1) were likely due to the overlapping electropherograms that generated a lot of confounding signals for base calling, and so, likely formed "chimeric sequences" (in addition to those potentially formed as a PCR-sequencing artifacts; (*Haas et al., 2011*) that far departed from any previously described organism. Contrariwise, the ≥98% identity with database entries found for three mixed sequences (Table 1) can be explained by a lower-intensity signal of the secondary (underneath) peaks, which did not significantly disturb the base calling process. Under these circumstances, such mixed electropherograms in several microbial studies (that have likely not been reported) may have been mistakenly overlooked or discarded as either poor sequencing or isolate contamination (*Chen et al., 2015*), when, in fact, they might have been indicative of intragenomic heterogeneity in 16S rDNAs.

A relevant result from our sampling of culturable endophytes from cacao was the high percentage of intragenomic variation for the 16S rRNA gene in this type of environmental community of culturable bacteria (Table 1). Out of the 65 isolates under study, at least 32 (49.2%) showed a mixed chromatogram coupled with an *Alu*I restriction pattern of ≥ 6 fragments, which safely detected more than a single template in the corresponding 799F/U1492R PCR amplicon. Considering the tropical origin of these isolates, this result suggests that part of the high levels of microbial diversity in these regions on Earth (*Arnold et al., 2002*; *Strobel & Daisy, 2003*; *Duarte et al., 2013*) is likely due to higher levels of genetic variability and gene transfer among bacteria (*Schouls, Schot & Jacobs, 2003*; *Zhaxybayeva et al., 2006*; *Jensen, Frost & Torsvik, 2009*; *Kitahara & Miyazaki, 2013*). An alternative possibility, though, that the culture medium used for isolation might have favored those bacteria with intrinsically higher rates of mutation/gene transfer cannot be ruled out. A common technical, though simplistic view for mixed chromatograms/sequences is that they result from contamination, due to certain associations of bacteria that are more difficult to separate by standard microbiological techniques, i.e., the case of syntrophs (*Caldwell, 1995*; *Sanders, 2012*). Nevertheless, such possibility is not expected to account for so many cases, mainly when very careful procedures for single-colonies isolation were employed throughout (see methods), and considering that endophytes do not tend to form syntrophic associations, due to the mostly aerobic growth conditions *in planta* (*Morris et al., 2013*). Therefore, it is fair to assume that contamination/syntrophy are likely not contributing to the results in a decisive manner (*Chen et al., 2015*). In addition, completely sequenced genomes in databases show that intragenomic 16S rDNA heterogeneity is not rare in *Bacteria*, with different levels and types of variation being taxon-dependent (*Sun et al., 2013*; *Chen et al., 2015*; *Valdivia-Anistro et al., 2016*); the same bacterial cell may have up to 15 copies of the 16S rRNA gene, and more than 2–3 types of sequences (Fig. 3) (*Klappenbach et al., 2001*; *Jensen, Frost & Torsvik, 2009*; *Liu et al., 2015*). Hence, such a condition of intragenomic 16s rDNA variation ought to be more careful addressed and considered, as it has direct consequences for species identification and analyses of richness, abundance and composition of bacterial communities in environmental samples (*Coenye & Vandamme, 2003*; *Pei et al., 2010*; *Tian et al., 2015*). Improved fitness to environmental stresses has been suggested as a driver for higher number and types of 16s rDNA in bacterial genomes (*López-López et al., 2007*; *Jensen, Frost & Torsvik, 2009*; *Chen*
*et al., 2015*), although these relationships are the likely result of a complex multi-factorial interaction (*Valdivia-Anistro et al., 2016*).

The approach here presented did show some limitations in identifying the 16S rDNA intragenomic heterogeneity. The fact that 36% of the mixed-sequences isolates showed a number of restriction fragments compatible to clean ones (Table 1) suggests that other interfering factors leading to mixed-type chromatograms have also occurred. Several technical aspects that can add experimental variability to chromatograms/restriction analyses are, for instance, (i) quality of the gel-purified PCR amplicon sent to sequencing; (ii) potential presence of non-specific priming sites; (iii) quality/consistency of the sequencing procedure, including signal strength, base-calling sensitivity and PCR-derived chimeric sequence formation (*Haas et al., 2011*); (iv) possible mutation(s) during PCR amplification or bacterial culture (*Martinez & Baquero, 2000*); (v) occasional V5–V9 regions with a single or none *Alu*I site that can lead to mixed chromatograms with ≤ 5 bands (*Ashby et al., 2007*); and (vi) occasional incomplete restriction digestion for certain sequences, as the conditions for full digestion (DNA and enzyme amounts, digestion time and temperature) vary among DNA structures/sequences and extracted biological samples. It is noteworthy that the efficiency/completeness of a type IIP enzyme digestion (such as *Alu*I) depends on specific sequences around the restriction sites and/or proximity between sites in a single DNA molecule (*Armstrong & Bauer, 1982*; *Alves et al., 1984*; *Pingoud & Jeltsch, 2001*). The latter condition is a possible explanation for the pattern of ≥ 6 bands obtained for one clean-electropherogram isolate (Table 1), since analysis of its V5–V9 sequence revealed two of the *Alu*I sites located less than 17 bp apart. Alternatively, other restriction enzymes can be used in the attempt to solve issues of this nature (e.g., *Stakenborg et al., 2005*; *Jensen, Frost & Torsvik, 2009*; *Dos Santos, 2017*). Once the technical aspects are addressed properly, the approach here employed can provide useful insights concerning the presence of 16S rDNA variation in natural populations at a first-tier level.

In the scope of ecology, diversity and evolution studies, certain aspects concerning intragenomic variability of 16S rDNA are important to highlight. It has been indicated that <97% identity between two 16S rRNA gene sequences is an indicative of different bacterial species (*Gevers et al., 2005*; *Janda & Abbott, 2007*). Interestingly, from our results, at least two different 16S rRNA gene sequences present in a single genome can account for differences higher than 3% identity (Figs. 2 and 3). Indeed, other authors have reported variations as high as 9.7% among the copies of 16S rDNA (e.g., *Pei et al., 2010*; *Sun et al., 2013*). This situation of cells containing heterologous 16S rRNA genes (*Asai et al., 1999*) is not so unexpected, as it can occur by horizontal transfer of complete operons between bacteria of the same or different species through transformation, conjugation and transduction (*Armougom & Raoult, 2009*; *Smillie et al., 2010*; *Arber, 2014*); studies have shown that *Bacillus* is a genus specially rich in terms of strains/cells harboring different types and copy number of rRNA operons (*Liu et al., 2015*), with such a variability being likely relevant for its wide niche occupancy (*Valdivia-Anistro et al., 2016* and references therein).

Therefore, analysis of unculturable populations from environmental samples through 16S rDNA sequencing may lead to a relevant overestimation of the community diversity

and composition (*Pei et al., 2010*; *Haas et al., 2011*; *Sun et al., 2013*; *Chen et al., 2015*), as different copies of this gene, although originating from the same cell, can match to different species in the database (Fig. 2). Our results suggest that assessments of microbial diversity from any environment, even if based upon high-throughput molecular techniques (e.g., *Sinclair et al., 2015*), may benefit from also obtaining culturable isolates to investigate the overall levels of their 16S rDNAs intragenomic heterogeneity. The concomitant use of a variety of different media (e.g., those focusing on culturing of specific bacterial subsets of interest; (*López-López et al., 2007*; *Jensen, Frost & Torsvik, 2009*; *Michon et al., 2010*; *Tchinda et al., 2016*) could help minimizing possible culturing biases of using a single-type medium, thereby providing more representative samplings. In this context, diversity overestimations of culture-independent methods can be somehow more precisely compensated. For instance, finding a given proportion of culturable isolates displaying such intragenomic variability (from a set of properly chosen media) could be applied to correct the OTUs diversity estimation found by a high-throughput molecular approach in the same sample. Depending on the previous knowledge existing in a study system, the information on OTUs overestimations provided by *Sun et al. (2013)* can be used with the overall approach presented here, which may further help in devising diversity-compensation strategies.

## CONCLUSIONS

In conclusion, the coupling of direct Sanger sequencing with restriction enzyme analysis on PCR-purified DNA from culturable isolates was capable to consistently reveal intragenomic 16S rDNA heterogeneity in endophytic culturable populations of bacteria. To our knowledge, this is a pioneer report assessing this issue in bacteria inhabiting cacao fruits/seeds, which can be viewed as a relevant representative of tropical environments for endophytes. Despite that full-genome sequencing is the most accurate way to verify such a intragenomic variability, this strategy depends on specific conditions of logistics or facilities for the laboratories worldwide. The approach presented here combined simple and classical procedures that can fast detect intragenomic variation in 16S rDNAs at a first-tier level in any bacterial population. This appears as an alternative not only for initial fingerprinting of isolates in culturable collections, but also to investigate microbial ecology and biodiversity in an array of environments.

## ACKNOWLEDGEMENTS

The authors are grateful to State University of Santa Cruz (UESC) for the infrastructure provided. The authors also wish to sincerely thank the anonymous reviewers who have contributed significantly to the improvement of the final version of the manuscript.

### Funding

The research was funded by the Brazilian National Council for Scientific and Technological Development (CNPq). Doctoral scholarship from the Bahia State Foundation for Research Support were granted to Cleiziane Bispo da Silva; two levels of scholarships from the Brazilian Coordination for the Improvement of Higher Education Personnel (CAPES, finance code 001) were granted to Hellen Ribeiro Martins dos Santos (doctoral), and to Ronaldo Costa Argôlo-Filho and Valter Cruz-Magalhães (post-doctoral). The funders had no role in study design, data collection and analysis, decision to publish, or preparation of the manuscript.

### Grant Disclosures

The following grant information was disclosed by the authors:
Brazilian National Council for Scientific and Technological Development (CNPq).
Bahia State Foundation for Research Support (FAPESB).
Brazilian Coordination for the Improvement of Higher Education Personnel (CAPES).

### Competing Interests

The authors declare there are no competing interests.

### Author Contributions

- Cleiziane Bispo da Silva, Hellen Ribeiro Martins dos Santos and Ronaldo Costa Argôlo-Filho conceived and designed the experiments, performed the experiments, analyzed the data, prepared figures and/or tables, authored or reviewed drafts of the paper, approved the final draft.
- Phellippe Arthur Santos Marbach and Jorge Teodoro de Souza conceived and designed the experiments, authored or reviewed drafts of the paper, approved the final draft.
- Valter Cruz-Magalhães conceived and designed the experiments, contributed reagents/materials/analysis tools, authored or reviewed drafts of the paper, approved the final draft.
- Leandro Lopes Loguercio conceived and designed the experiments, analyzed the data, contributed reagents/materials/analysis tools, prepared figures and/or tables, authored or reviewed drafts of the paper, approved the final draft.

### Data Availability

The raw data is available as Supplemental Files.

### Supplemental Information

Supplemental information for this article can be found online at http://dx.doi.org/10.7717/peerj.7452#supplemental-information.

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
