# Peer review of "First-tier detection of intragenomic 16S rRNA gene variation in culturable endophytic bacteria from cacao seeds"

_PeerJ, doi:10.7717/peerj.7452_

## Round 0.1 · original submission · Major Revisions

The reviewers commented very favorably on the manuscript and on the important of the work. However, a number of concerns were raised about the methodology and the resulting validity of the results. As such, major revisions are necessary.

Reviewer 1 ·

Basic reporting

The manuscript was clearly written with clear language and sufficient background and references. Figures are relevant and appropriately described. Although not strictly required, I recommend that the raw electropherograms produced for each culture should be made available in order to facilitate reanalysis and independent verification.

Experimental design

The research question was well-defined and the methods were described in sufficient detail to replicate the work. Unfortunately, the overall experimental design falls short of the necessary rigor to reach the authors’ conclusions.

It is clear that the authors have detected heterogeneity among the 16S rRNA gene sequences that can be amplified from their cultures. However, it was never shown that these cultures represent pure isolates. It is likely that they are instead enrichments of multiple organisms and that the heterogeneity observed can be attributed to intergenomic and not intragenomic heterogeneity. Three round of streak plating (lines 140-142) is not sufficient to establish a culture as being a pure isolate. Given that direct amplification of 16S rRNA genes is often used as a purity test, the obvious interpretation is that they do not have pure isolates. Therefore, the burden of proof lies on the authors to reasonably prove that their cultures are pure, more precisely, the authors must prove that their cultures were pure when they were used for generating the electropherograms and Alu1 resctriction patterns found in the manuscript. Unfortunately, it is quite difficult to recommend an alternative method of verifying purity of the cultures. Cloning 16S rRNA genes amplified from cultures would only verify heterogeneity and would not address the question of purity. Fluorescent in-situ hybridization (FISH) can be used to verify purity, but this procedure requires extensive experience and well-characterized primers, which if either too specific or two broad could easily lead to a misinterpretation. The only other alternative I can think of would be either deep sequencing with short reads (e.g. Illumina) to rule out potential contaminants or alternatively to sequence their cultures with long-read technology (e.g. Nanopore) to directly verify intragenomic rRNA gene heterogeneity.

Validity of the findings

Given the major problem identified with the experimental design (see section 2), the validity of the findings are seriously in question.

Additional comments

I appreciate the time and effort put into this manuscript and I appreciate how important this work is. The manuscript is clearly well organized and clearly written, which made it a pleasure to review.

Reviewer 2 ·

Basic reporting

In their manuscript, authors B da Silva et al., propose a rapid, cost effective method for the detection of intragenomic 16S rRNA gene variation in culturable bacteria endoseeds from cacao plants. In general good English language is used throughout the manuscript (see minor specific comments). Introduction section in well presented and the literature is well referenced and relevant. Finally all figures are under PeerJ’s standards and the raw data are supplied.

Experimental design

Methods described with sufficient detail & information to replicate. More comments are included in the following section because they are a combination of used methods and obtained results.

Validity of the findings

The study deals with a well known problem for all diversity studies, no matter the ecosystem, the overestimation or not of the monitored diversity based on NGS results on the 16SrRNA gene. It is well documented that this problem is due to 1. multiple copy numbers of 16SrRNA gene within a cell, 2. horizontal gene transfer and 3. intragenomic 16S rRNA gene variation of the multiple copy numbers of 16SrRNA gene. Whole genome sequence at the single-cell level is a solution, for those bacteria that can be cultivated and for those who can afford the cost of this method. Combining culture dependant and dependant methods is a partial solution tho this problem, with the limitations described by the authors. In fact the authors have consider and reported all methodological limitations and drawbacks.
However, there is a lack in commenting on the hypervariable region of the 16SrRNA the authors targeted (V5-V9). There is a very interesting review of Sun et al. (2013), that is referenced by the authors, that highlights the region V4-V5 as with the less intragenomic variation. Could the authors please explain why they used V5-V9 region and discuss that according to their results. Do they believe that the region they target could lead to the patterns they monitor?
In addition, in my opinion, a more clear statement should be made on how this kind of analysis can help on assessing a more accurate diversity pattern of the ecosystem under study. The authors make an interesting remark on the introduction (L99-L101: “..to provide correction factors to compensate it”), however they don’t further discuss this in their manuscript.
Finally, as a comment, this kind of analysis could be a useful first finger-print for the expected diversity or identification of isolating the same bug multiple times, especially when you want to continue your analysis with searching for e.g. plant growth promoting traits for your isolates.

Additional comments

Minor comments:
1.L73 “ Bacterial strains with > 70% at the DNA reassociation level”: maybe to replace with DNA-DNA reassociation
2.L229-232: This sentence could be re-written in more simple way
3.L232 “ a single PCR amplicon”: do you mean single gel band? Could you please rephrase?
4.L244 “On the other hand”: This is not needed
5.L245: It is not clear for which electropherograms you performed in silico analysis
6.L345-350: Did you examined if there was any correlation in intragenomic variability at lower taxonomic level, eg family or genus of the full genomes used with the isolates tested?

---

## Round 0.2 · accepted · Accept

Thank you for your detailed rebuttal letter and careful revisions. I believe you have sufficiently addressed all of the reviewer's concerns about the purity of the cultures.